# Staff, resident, and care partner perceptions on the use of a personalized tablet to mitigate the impact of isolation in long-term care residents

Kristina M. Kokorelias[1,2,3,4], Alisa Grigorovich[5,6], Teresa D'Elia[5], Arlene Astell[2,4,5], Josephine McMurray[7], Andrea Iaboni[5,8]*

**1** Section of Geriatric Medicine, Department of Medicine, Sinai Health System and University Health Network, Toronto, Ontario, Canada, **2** Department of Occupational Science & Occupational Therapy, Temerty Faculty of Medicine, University of Toronto, Toronto, Ontario, Canada, **3** National Institute on Ageing, Toronto Metropolitan University, Toronto, Ontario, Canada, **4** Rehabilitation Sciences Institute, Temerty Faculty of Medicine, University of Toronto, Toronto, Ontario, Canada, **5** KITE Research Institute, Toronto Rehabilitation Institute – University Health Network, Toronto, Ontario, Canada, **6** Recreation and Leisure Studies, Brock University, St. Catharines, Ontario, Canada, **7** Lazaridis School of Business & Economics/Community Health, Wilfrid Laurier University, Waterloo, Ontario, Canada **8** Department of Psychiatry, Temerty Faculty of Medicine, University of Toronto, Toronto, Ontario, Canada,

* andrea.iaboni@uhn.ca

## Abstract

### Background and Objectives

The Dementia Isolation Toolkit (DIT) project developed DIT-Tech, a tablet-based tool to engage residents. Implementing such technology faces challenges like digital literacy and organizational resistance. This study aimed to develop an understanding of the staff, resident and care partner experiences, including barriers and facilitators to the adoption of remote-access personal tablets in long-term care homes (LTCHs).

### Research Design and Methods

Guided by the FITT framework, which emphasizes the alignment between technology, users, and clinical activities, this investigation sought to uncover the obstacles and drivers influencing the integration of DIT-Tech within the LTCH setting. Recruitment involved voluntary participation of various stakeholders within the LTCHs: 20 staff members, 23 care partners, and 7 residents who received the DIT-Tech tablets. Purposeful selection ensured representation across demographics and levels of tablet usage. Over the research period, a total of 59 in-depth interviews were conducted via telephone or video calls. Data collection and analysis occurred simultaneously. Coding strategies, incorporating both inductive and deductive approaches, were employed.

### Results

The study highlighted pivotal factors in DIT-Tech implementation within LTCHs. Initial enthusiasm among staff and care partners was countered by staff resistance due to

**Data availability statement:** Our de-identified data set contains potentially sensitive patient information. Due to the risk of re-identification and the ethical considerations surrounding the sharing of health-related data, data access is restricted. These restrictions have been imposed by the University Health Network Research Ethics Board (REB), which has approved the study under specific data privacy and confidentiality guidelines. All data access queries can be addressed to Leia Shum MASc. Biomedical Engineering Research Associate (Toronto Rehabilitation Institute - KITE) at leia.shum@uhn.ca, who is the research associate who provides data management support for our research team.

**Funding:** This research was funded by a Province of Ontario Rapid Access COVID-19 research grant. It was also supported by the AGE-WELL Network of Centres of Excellent (NCE) and the Walter and Maria Schroeder Institute for Brain Innovation and Recovery.

**Competing interests:** The authors have declared that no competing interests exist.

workload and past tech issues. Personalization benefited residents but usability challenges and poor integration posed barriers. Aligning tech with organizational goals is crucial. Privacy measures were valued. Care partners and residents embraced DIT-Tech, emphasizing the need for targeted support for staff.

## Discussion and Implications

These findings stress the necessity of robust guidelines for implementing remote-controlled tablets in LTCHs, providing vital insights for enhancing technology in care settings.

## Introduction

The COVID-19 pandemic severely impacted the mental health and well-being of long-term care home (LTCH) residents and those who care for them (e.g., care staff, family) [1]. Residents faced repeated outbreaks in their LTCH, resulting in the need to isolate for long periods alone in their rooms (henceforth referred to as 'isolation') [2]. Isolation, which refers to reduced social contacts and interactions, is associated with loneliness [3] and mental health problems, particularly, anxiety and depression [4]. Loneliness has long been a recognized issue in LTCHs [5], particularly among residents without cognitive impairment [6]. However, the enforced isolation due to COVID-19 exacerbated loneliness and deteriorated mental health [7]. Residents' experiences of confinement and separation from their care partners and peers were compounded by limited activity or sensory stimulation. Infectious disease outbreaks continue to be frequent events within LTCH, and it is crucial to develop innovative approaches to meet the needs of residents during isolation. One approach to this challenge involves the use of technology to engage residents in activities and more broadly support their mental and physical health [8]. Playing digital games on tablets is an enjoyable activity for older adults, including those living with dementia [9]. Tablets can also be used for video-calling and staying connected with families, but availability and awareness of their potential are limited within LTCH [10].

The Dementia Isolation Toolkit (DIT) project (www.dementiaisolationtoolkit.com), developed DIT-Tech, aimed at meeting residents' social and recreational needs during isolation [11,12]. This tablet-based tool assists recreation staff in engaging residents without physical visits and enables video communication with families. Tailored to individual preferences, DIT-Tech offers personalized interfaces and adaptive features. Successful technology-based interventions in LTCHs hinge on multiple factors [13], encompassing technology nature, resource availability, integration strategies, work culture, staff challenges, and regulatory environments [14]. Barriers like technological complexity, digital literacy gaps, and resistance to change hinder implementation [15]. Comprehensive guidelines for resident-focused technologies in LTCHs, like tablet-based interventions, are currently lacking. Establishing implementation procedures is crucial for consistent, effective interventions and successful outcomes [16].

We conducted a mixed methods study of the DIT-Tech involving 57 resident-care partner pairs and evaluated factors influencing tablet use, as well as the impact on social engagement and quality of life [17]. Data were collected through surveys, interviews, and tablet usage tracking, with findings suggesting the potential of personalized technology to address isolation in LTCHs. The primary outcomes of this study highlighted variability in use, influenced by cognitive abilities, sensory impairments, and caregiver proximity, with customized approaches proving critical [17].

This qualitative study utilizes interviews conducted during the broader DIT-Tech study to help understand the practical use, challenges, and enabling factors of DIT-Tech in LTCH, and to offer insights for facilities and researchers aiming to implement similar technologies. Studies exploring effective technology-based interventions for older adults should prioritize contextual "fit" across diverse settings [18]. Customized implementation strategies, responsive to specific contexts, increase sustained success likelihood. The ("**F** it between **I** ndividuals, **T** ask and **T** echnology") (FITT) framework emphasizes aligning user attributes, technological features, and clinical tasks for success in LTCH settings [18]. The FITT framework, suggests that technology-based interventions in LTCH settings should consider how user attributes (such as computer anxiety and motivation), technological features (like usability and functionality), and clinical tasks (including organizational elements) align for success [18]. This model aids in examining socio-organizational and technical aspects contributing to technology adoption's success. Leveraging the FITT framework to analyze technological impacts via LTCH staff, residents, and care partners' perspectives provides comprehensive insights and areas for enhancement.

To achieve the aim of understanding adoption challenges and benefits, this qualitative study explored staff, resident, and care partner experiences with remote-access tablets in 7 LTCHs. Using the FITT framework, interviews assessed the impact on residents and care partners, offering insights into effectiveness and areas for enhancement in technological interventions within LTCHs.

## Methods

### Design

This qualitative descriptive study [19] used purposive sampling to recruit staff members, residents and care partners from multiple long-term care homes (LTCH) involved in the DIT-Tech study. Full details of the DIT-Tech study and participants and the analysis of the study primary outcomes can be found in [17]. In this qualitative aim of the study, data was collected through virtual semi-structured interviews conducted, to gather in-depth perspectives on the adoption of the DIT-Tech tablet. Thematic analysis was employed to identify key barriers and facilitators, with coding done using NVivo 14 software to ensure a systematic approach. This methodology allowed us to capture diverse experiences and contextual factors influencing the success of the intervention. This approach allows for a clear, practical understanding of real-world implementation challenges and successes, making it suitable for informing future interventions and practice improvements [20]. The University Health Network Research Ethics Board (REB: 21–5535) reviewed and approved all research activities.

### Theory

This study was informed by the FITT framework [18], which conceptualizes the adoption of technology in a clinical environment (e.g., LTCH) as a 'fit' between users of the technology, the technology itself, and the clinical activities it is designed to support [18].

### Setting and intervention

This study was conducted in 7 LTCHs in Ontario, with each receiving 10 tablets: 9 for residents, 1 for a "DIT-Tech Coordinator." The recruitment of LTCHs involved a strategic approach, wherein potential sites were identified based on existing collaborations with the research team and their willingness to participate. The long-term care homes represented urban and rural settings and cared for residents of various socioeconomic backgrounds, and with varied resident needs. The

diversity of the LTCHs was characterized in terms of representation of different regions of Ontario and size of community. Of the 7 homes, 2 were located in a large city, 3 were located in different small cities (50–150,000 people), and 2 were located in small towns (5–15,000 people). 5 were operated by not-for-profit organizations and 2 were operated by a municipality. In terms of size of the homes, they ranged between 106–228 beds per home, with an average size of 142 beds. This diversity was crucial to capture a broad spectrum of perspectives and experiences regarding the implementation of the DIT-Tech tablet intervention. These Android tablets, property of the LTCHs, feature standard apps and Skype. Key features include remote control access, direct family calls via auto-answer on Skype, and personalized setups for residents [17].

Full study details on the implementation of the tablet are available in the primary study [17]. Briefly, tablets were provided to a single resident for their personal use, ensuring that the technology was tailored to their individual needs and preferences. The personalization of each tablet was a collaborative effort that involved information gathered from both residents and their care partners. This process included discussions with care staff aimed at understanding the residents' interests, preferred communication methods, and any specific applications that would enhance their engagement. If a tablet was to be reassigned to another resident, it would be reprogrammed to reflect the new resident's preferences and requirements. The responsibility for individualizing the tablets primarily lay with the staff, who worked closely with the research team to implement the necessary customizations based on resident input and care partner feedback. The staff members interviewed played a significant role in supporting residents' use of the tablets. They assisted with training, troubleshooting, and ongoing encouragement for residents to engage with the technology.

In our study, "tablet usage level" refers to the frequency and extent of interaction residents had with the tablets, which can include the duration of use, types of applications accessed, and engagement in activities facilitated by the tablets. The median tablet use by participants was 7 minutes per day [17].

## Participants and recruitment

From the 58 residents/care partner dyads who received tablets, 23 care partners were interviewed, and 7 residents participated in interviews with their care partner. Care partners were purposively selected from those who indicated a willingness to participate in an interview, to ensure diversity in terms of demographics, caregiving relationship, and those with more direct and indirect involvement in the residents' lives.Twenty staff were also recruited, based on capture a range of experiences, roles, and interactions with the DIT-Tech tablet. The goal was to include participants from varied professional backgrounds, including nursing, recreational therapy, and social work, to reflect different perspectives on the tablet's implementation. All residents who provided consent or assent to participating in an interview, and where there was agreement of their participating care partner to also participate, were included. Interviews were completed as dyads to have the support of the caregiver in facilitating the virtual visit and to support inclusion of residents with cognitive impairment.

## Data collection

Between September 2022 and July 2023, 59 telephone/video interviews were conducted by a qualitative research coordinator. Staff interviews (n = 20) preceded implementation, followed by post-implementation follow-ups (n = 16 staff, n = 16 care partners, n = 7 resident/care partner dyads). The FITT-based interview guide (S1 Appendix) explored individual, interactional, task, technological, and temporal aspects. Audio-recorded interviews were transcribed professionally, anonymized, and verified by the research team without sharing with participants.

## Data analysis

Data were collected and reviewed simultaneously. Inductive and deductive approaches to qualitative thematic analysis were used [21]. Specifically, each week, the research coordinator would share details of the interviews with the research team. Next, the first set of transcripts were open coded independently by KMK (a PhD-trained female researcher) and

TD (a MA-trained female researcher) using coding strategies proposed by Quresh and Ünlü [22]. These authors then met to discuss their coding and these discussions formed the development of an initial coding framework, which provided a more focused understanding of participant experiences. Codes representing the FITT framework were added to the open-coding scheme to deductively guide the coding of the interview data. This broad framework was shared with and confirmed by the larger research team [18]. The coding framework was further refined as additional interviews were reviewed, as well as through elements of the framework method [23]. For example, we used the code "motivation" to delineate attributes of the participants and "usability" to note attributes of the technology. As our analytic understanding of each LTCH site emerged and knowledge from the interviews developed, we returned to the transcripts to recode as necessary.

The coding framework, established and applied using NVivo 12 software [24], underwent team review across all transcripts. Meetings led to grouping codes into emerging themes, examining LTCH data for variations. Themes were refined iteratively to grasp implementation insights and outcomes. Differences pre and post-study were noted. Rigorous analysis involved multiple individuals, fostering analytic rigor and consensus on final themes presented below, enhanced through triangulation and team meetings [25].

## Results

The demographic characteristics of the full study sample are available in the larger mixed-methods study [17]. Interviews with a subset of 7 LTCH residents, 23 care partners and 19 staff are included in this study. See Tables 1–3 for care partner, resident, and staff interview participant characteristics, respectively. Several key themes emerged: first, the interplay of motivation and resistance was a vital determinant of successful implementation, underscoring the central role of personal attitudes and determination. The personalization of DIT-Tech emerged as a key facilitator, enhancing the adaptability to the specific needs and interests of residents. Additionally, system usability, the presence of support, and the alignment of technology with existing workflows, were identified as critical factors in either hindering or promoting adoption. Excerpts from interviews are used to illustrate the themes and are identified by participant ID, participant role, LTCH ID, gender and age. Notably, resident quotes were largely omitted from the excerpts, as they frequently echoed sentiments expressed by their care partners with brief replies. However, there were no instances of contradictory responses between dyad participants.

We identified three major themes with eight subthemes related to participant experiences implementing and using the DIT-Tech. Themes represent the interaction between FITT constructs 1) individual and task: subthemes motivation, resistance, 2) individual and technology: subthemes personalization, usability, and 3) task and technology: subthemes workflow integration, technology support, poor management/technology/organizational fit. Within each of these, we describe the themes as barriers and/or facilitators. Our analysis suggests this was an effective intervention from the perspective of care partners and residents but less so from the perspective of staff. Excerpts from interviews are used to illustrate the themes and are identified by participant ID, participant role, gender and age.

### Individual and task

Motivation and resistance emerged as key themes in the study, reflecting the attitudes, behaviours, and responses of individuals involved in the task of implementing DIT-Tech. These themes highlight the influential role of motivation in driving successful implementation and the significance of resilience in overcoming challenges. Additionally, we identified increased workflow tied to resistance as a barrier to effective outcomes.

**Motivation.** The use of DIT-Tech was initially significantly facilitated by the motivation of staff and family members for supporting residents' use particularly to enhance their engagement in leisure activities. Prior to the implementation of the tablets, resident, staff and family member participants noted that the remote access and auto-answer video call features differentiated DIT-Tech from other technologies and thus, were excited about their use. They perceived the tablets, especially the games on these, as promoting cognitive stimulation, social engagement, and increasing resident enjoyment. One care partner noted

**Table 1.  Care partner participant characteristics.**

| CARE PARTNER INTERVIEW PARTICIPANTS (n=23) | | Total |
|---|---|---|
| **Demographics** | | **N=22** |
| **Age** | Mean (STD) | 63.8 (11.7) |
| | Range | 36-64 |
| **Gender** | Female | 18 (82%) |
| | Male | 4 (18%) |
| **Relationship** | Spouse | 4 (18%) |
| | Sibling | 4 (18%) |
| | Child | 10 (46%) |
| | Other (Grandchild, Daughter-in-Law, Sister-in-Law, Niece) | 4 (18%) |
| **Education** | Secondary | 4 (18%) |
| | University, College, Post-secondary | 11 (50%) |
| | Post-graduate | 5 (23%) |
| | Other | 2 (9%) |
| **Distance** | Same neighbourhood | 4 (18%) |
| | Under 10 km | 8 (36%) |
| | Less than 25 km | 6 (27%) |
| | 25-100 km | 1 (5%) |
| | More than 100 km | 3 (14%) |
| **Primary Caregiver** | Yes | 12 (55%) |
| | No | 10 (45%) |

**Table 2.  Resident participant characteristics.**

| RESIDENT INTERVIEW PARTICIPANTS (n=7) | | Total |
|---|---|---|
| **Demographics** | | **N=7** |
| **Age** | Mean (STD) | 79.5 (8.9) |
| | Range | 63-87 |
| **Gender** | Female | 3 (43%) |
| | Male | 4 (57%) |
| **Ethnicity** | White | 6 (86%) |
| | Non-White | 1 (14%) |
| **Marital Status** | Married | 4 (57%) |
| | Never Married | 2 (29%) |
| | Divorced | 1 (14%) |
| **Cognitive Performance Scale (CPS) Score** | 0 | 2 (28%) |
| | 1 | 0 |
| | 2 | 1 (14%) |
| | 3 | 4 (57%) |
| | 4 | 0 |
| | 5 | 0 |

**Table 3. Staff participant characteristics.**

| STAFF INTERVIEW PARTICIPANTS (n = 20) | | Total |
|---|---|---|
| **Demographics** | | **N = 19** |
| **Age** | Mean (STD) | 34.2 (11.5) |
| | Range | 20-58 |
| **Gender** | Female | 17 (90%) |
| | Male | 2 (10%) |
| **Ethnicity** | White | 15 (80%) |
| | Non-white | 4 (21%) |
| **Education** | University, College, Post-secondary | 17 (90%) |
| | Post-graduate | 2 (10%) |
| **Qualifications** | Personal Support Worker (PSW)/ Home Care Assistant (HCA) *Typically require a certificate from a recognized training program. Programs usually last between 6–12 months and focus on basic personal care, communication skills, and support for daily living activities.* | 6 (32%) |
| | Registered Practical Nurse (RPN) *RPNs hold a diploma in practical nursing, which typically involves 2–3 years of education and clinical training. They must also be licensed by the provincial regulatory body to practice.* | 2 (10%) |
| | Registered Nurse (RN)/ Bachelor of Science in Nursing (BScN)/ Bachelor of Nursing (BN): RNs must have completed a Bachelor of Science in Nursing (BScN) or an equivalent degree, which typically requires 4 years of university education. They must also pass a national licensing examination and be registered with the provincial nursing regulatory body. | 3 (16%) |
| | Other | 8 (42%) |
| **Role** (select all that apply) | RN | 2 (10%) |
| | PSW | 5 (26%) |
| | Recreation Staff: Often hold a degree or diploma in recreation therapy, gerontology, or a related field. Certification in therapeutic recreation may also be required or preferred. | 11 (58%) |
| | Behavioral Supports Ontario (BSO) Staff *Typically include professionals with backgrounds in nursing, social work, psychology, or gerontology. Specialized training in behavioral management and support for individuals with complex needs is often required.* | 1 (5%) |
| | Other (Researcher, PRC, Recreation Programmer) | 3 (16%) |
| **Experience** | Less than 5 years | 10 (53%) |
| | 6-10 years | 3 (16%) |
| | 11-15 years | 3 (16%) |
| | 16-20 years | 1 (5%) |
| | More than 20 years | 2 (10%) |

*"Presence is important, or at least things to keep her active are quite important to her current state, her behavioral state. That's where the tablet actually is very handy"* (ID I-09, Care Partner, Spouse, Male, 64).

Following distribution of the tablets, staff members and family care partners were highly motivated by the ability of DIT-Tech to offer personalized experiences to residents based on individual interests. For example, games and applications, as well as music and video playlists, were downloaded based on individual preferences. The tablets also supported various functions that met resident needs, such as communication tools (e.g., Skype, WhatsApp), and access to personalized content (e.g., films, sports, and news). One staff member noted, "*For [one resident] it was so nice to just have that tablet., He can set himself up, do whatever he wants on it, which is nice.*" (ID C-01, Staff, Female, 22) Thus, staff and care partner participants were motivated by their own desire to improve the well-being and quality of life of LTCH residents. They perceived DIT-Tech as a supplement to caregiving routines and recreation efforts. Care partners in particular expressed

gratitude for how DIT-Tech provided opportunities for families to engage in activities together, highlighting the value of intergenerational connections facilitated by advanced technology:

> "*Well, I like to see my son and him doing something together on it. That's been great. I'd like to thank the team for that, for offering that kind of more advanced technology to a group of folks who maybe wouldn't get to access it. I think that intergenerational bridging, that's been great*" (ID D-09, Care Partner, Child, Female, 47)

**Resistance.**  Staff resistance was identified as a key barrier to implementation prior to the tablet starting. Many LTCH staff were not interested in using new technology. Their reluctance was due, in part, to past negative experiences with technologies that increased their workload. One staff participant recounted assisting residents with their personal technology and noted,

> "*It can get frustrating to use them as an older generation and everything and understanding because I know it takes some time [...] when they have the questions, just to explain it and go through it with them patiently*" (ID D-01, Staff, Female, age n/a).

Moreover, some staff were hesitant to deviate from familiar methods and routines. This resistance stemmed from past use of technology that was ineffective in supporting the leisure activities of residents, or beliefs that older adults cannot, or choose not to, use technology. One of the participants reported:, "*I heard a staff member say, "Why do they have this? They're [residents] not going to use it*" (ID C-02, Staff, Female, 33). Similar ageist perspectives regarding older adults and technology were notable in several staff and care partner interviews.

Prior to the implementation period, some staff and care partner participants reported limited experience and confidence in using tablets, highlighting their lack of technological literacy as a primary factor contributing to their resistance. They perceived tablets as complex devices that required additional training and support. The increased workload associated with introducing new technology and the associated activities, was a particular concern expressed during follow up interviews with participants working in homes that were coping with staff shortages. However, in facilities with designated resources for technology support (e.g., designated IT staff roles, technology savvy recreation staff), resistance was considerably lower. One staff participant shared that where there is no formal technological support:

> "*It's going to be a little bit of a struggle, not understanding what's technically wrong with the technology itself. If something is defective or anything like that, just trying everything and making sure that we've done everything to make sure that it's troubleshooted and it's working properly.*" (ID D-01, Staff, Female, age n/a).

Some participants continued to express doubts about the effectiveness of tablets in enhancing communication and engagement with residents during the study period. This skepticism played a part in the resistance and reluctance to adopt this technology. One staff participant noted:

> "*The first instinct from [residents with cognitive impairment] using it, they find it a bit difficult understanding because the interface is very different for them. They're not used to it, and most of the residents haven't really actually had computers in their previous lives, so them using a tablet is like the next step beyond a computer, and that can be very confusing*" (ID I-03, Staff, Male, 35)

Some staff described being particularly overwhelmed by existing clinical and administrative duties before the introduction of DIT-Tech. Learning to use a new technology was seen as adding to their workloads and as a result, this led to lower

use. Moreover, some staff users felt their workload was higher than that of non-users, without appropriate compensation. One participant explained,

> "*The negatives are that the staff always have to be there to operate it for them. Like, I don't think any of them, I can't think of a single resident that could operate any of the technology themselves.*" (ID E-02, Staff, Female, 35)

The demands of assisting multiple residents with the tablets, coupled with the need to address technology-related challenges during a shift, contributed to staff time constraints and limited their availability to fully utilize DIT-Tech. These issues were sometimes compounded by a lack of hands-on training, resulting in staff expressing frustration with colleagues and contributing to a negative perception of the technology. One staff participant described:

> *From my knowledge, the staff members, when you bring new technology stuff in, they always find it harder. They have to learn all these new rules and how to use it, and they sometimes find-- These tablets, for example, are smaller, and they can easily get lost if you're not careful, and they have to learn a whole new range of how they interface with it and whatnot* (ID I-03, Staff, Male, 35).

### Individual and technology

The primary barriers to implementation were related to a poor fit between individual users (residents and care partners) and the technology. Factors that facilitated the implementation included user-friendliness and the ability of the tablets to be personalized to meet users' recreational needs. Staff described issues with usability.

**Personalization to meet recreational needs.** Staff participants noted during both the pre-intervention and follow up interviews that the ability to personalize the features of DIT-Tech improved its overall acceptance among users. Providing each resident with their own personal device, allowed staff to modify the recreational activities for each resident by selecting activities tailored to their individual preferences. This adaptability supported various communication methods (e.g., talk to text), making it easier for staff to support tablet use. Moreover, the personalizing of tablets was perceived as improving workflow efficiency, by equipping the tablets with applications and tools that matched residents' needs. This was perceived as helping to overcome administrative burdens and enhance time management, allowing staff to allocate more time to resident recreational routines. On the other hand, for some staff, the need to download specific applications was perceived as increasing their workload.

Similarly, care partners noted benefits from the personalization of the tablets, providing examples of tailored leisure activities and content that aligned with the interests and preferences of their care recipient. One care partner explained,

> "*I've loaded a Bell Fibe app on it, and I can put her beloved Blue Jays on I-- watch the Blue Jays, or a tennis match, or things that she was really active in, that still strikes a memory chord with her and that she really enjoys escaping in that way.*" (IDI-09 Care Partner, Spouse, Male,64).

Some care partners described incorporating video chat applications used by multiple family members and friends, such as WhatsApp and Facebook Messenger, to help residents maintain and expand their social networks. This offered the convenience of audio/ video calls and chatting without having to rely on family or staff. This was particularly valuable for residents with limited English proficiency, financial constraints for long-distance calls, or barriers to communication within the LTCH (e.g., unable to hold a phone).For example:

> "*With the tablet, you are able to also have a conversation for more than 5, 10 minutes. Maybe half an hour, 45 minutes. That's also a huge difference because when you call long distance it's also a matter of how much you're paying for that long distance call*" (ID I-08, Care Partner, Sibling, Female, age n/a).

Care partners encouraged activities that provided cognitive stimulation, selecting and personalizing applications that challenge memory, problem-solving or creativity. This personalization, including access to preferred music, religious services, and first language content, empowered residents by offering them a sense of ownership over the tablet. Care partners and residents reported having a "personal" device was highly valued as it did not require sharing with other residents. One care partner shared:

*"That's a big deal. What's set out to the TV is not interesting. [Resident is] a little unique in that she's only just turned 60. The content that might appease somebody in their 80s or 90s is not the same content that interests her. Being able to personalize and give her sporting events, which is what she'd always liked to watch, is a big deal"* (ID I-09, Care Partner, Spouse, Male, 64).

Likewise, one resident shared that they appreciated the fact that they can choose their favourite movies to watch, rather than just watching the ones the LTCH provides:

*"I'm pretty sure that [residents] watch it all the time. They watch that movie every day, the English Fellow. Anyways, you can get to choose the movies. They love the movies. It keeps someone busy while they're in isolation"* (ID G-09, Resident, Female, 70).

**Usability.** The usability and ease of use of the technology were crucial factors in its implementation and depended on the prior experience of staff, care partners and residents with tablet devices. A few participants noted that the various applications on the tablet posed significant barriers to adoption. Barriers included counterintuitive navigation for downloading new applications, and unclear instructions for troubleshooting from the LTCH and the DiT-Tech research team. Despite some training organized by management and the research team to familiarize staff and care partners with the system, certain resident participants struggled to use DIT-Tech. One care partner said:

*"He [resident] was an engineer back in the day. He loves to know how things work and especially with the dawn of the internet and all of that in the last couple of decades in particular, he was quite interested in trying to master it and be part of it and all of that. At this point in his life, it just was a little bit too difficult."* (ID I-06, Care Partner, Child, Female, 66).

Conversely, for participants with previous tablet experience, features like simple icons, remote control and interactions were perceived as beneficial, facilitating adoption of the new system. One care partner noted the perceived ease of use from the resident's perspective:

*"I guess we were wondering its ease of use and all of that, but as it turns out, from his point of view, [the dial-in and remote-control elements] were very passive"* (ID I-06, Care Partner, Child Female, 66).

Participants who found DIT-Tech user-friendly noted intuitive controls, and clear instructions, that helped them quickly learn and become proficient in its use.

Care partners highlighted several distinctive features of DIT-Tech, including the large 10-inch screen, which helped residents with visual impairments in using the tablet more autonomously, and engage more than with smaller smartphone screens or audio-only devices. The protective rubber case, which could be used as a stand, and the tablet's portability, allowing residents to use it beyond their rooms, were also appreciated. For residents with difficulties using the touch-screen (such as those with hand tremors), stylus pens were made available upon request, and the Google Assist voice command feature proved invaluable in such instances. This voice-command functionality enabled some residents to

independently access content on platforms like YouTube and search for online materials, enhancing their independence and engagement with the technology.

**Task-technology**

Task-technology fit required aligning DIT-Tech with existing work processes. Utilizing formal or informal technology support and aligning the use of the tablets into standard practices were identified by participants as necessary steps. However, the lack of organizational and management support during implementation created barriers to implementation and usage.

**Integration Into workflow processes.** During the follow up interviews, some staff described incorporating the tablets into their daily routines through regular use and making a conscious effort to discuss DIT-Tech with family and residents to familiarize them with the technology. One staff participant noted:

*"What I found was that trying to schedule the tablet into the full day was tricky because you have to sit with the resident for X amount of time to get them to fully test it. I had to set aside time throughout the week regularly so they [resident] would get familiar with it because they were having a hard time getting used to it. Constantly seeing them every single day or every couple of days just to really get used to be the extra step, which was hard to implement into regular whole day of activity programming."* (ID I-03, Staff, Male, 35)

Despite their efforts, care partners and residents often did not want to ask staff to help them use the technology due to perceived time constraints experienced by staff and their desire to not further burden them. One care partner explained: "*I wouldn't want to per se impose that on the staff because I know that the staff has so many things to do with all the other residents*" (ID I-08, Care Partner, Sibling, Female, age n/a)

To mitigate this, staff sometimes pre-loaded the tablets with personalized playlists, videos, or interactive apps and requested volunteers or family members to spend quality time with residents using the tablets. This collaborative approach allowed staff to focus on their primary care responsibilities while ensuring that residents had access to leisure opportunities facilitated through the DIT-Tech.

**Technology support.** The presence of technical support staff during working hours played a crucial role in addressing system-related issues such as connecting to the internet and downloading new applications promptly. Furthermore, staff, residents and care partners reported broader factors like unstable Wi-Fi quality and the influence of online security firewalls, significantly impacted the accessibility and usability of DIT-Tech. Staff had access to technical support from the research team through group meetings, emails and telephone calls as needed. In some LTCHs, informal technology support was provided by staff members who were more familiar with technology and assisting others in system use. This fostered a sense of reliability and prompt resolution of technical problems and requests. One staff member noted:

*"I know I had one tablet that was having problems with opening any of the programming by itself. When I couldn't get it to work, I just called [colleague] to come down"* (ID D-01, Staff, Female, age n/a)

Care partner participants benefitted from LTCH staff support with DIT-Tech, as staff support minimized downtime and frustration for themselves and residents. It empowered care partners to explore the tablet functionalities, experiment with different features, and gradually develop their skills and familiarity with the device to support resident use. Staff and care partners also highlighted the value of the research team's ability to remotely access the tablets to provide prompt and effective support and troubleshooting.

**Poor fit between the organization and management and the technology.** Staff believed the biggest barrier to implementation was misalignment with the overall goals, structure, and management practices of the LTCHs. Staff reported frustration as they did not think management thoroughly considered their needs, resources, and capabilities when deciding to implement DIT-Tech. One participant noted that the culture of LTCHs was,

*"Trying to pack in as many things as you can for them [residents] as possible during your day"* (ID C--03, Staff, Female, 55).

Frontline staff noted that insufficient communication and collaboration between the technology and recreation departments and other organizational units, hindered the integration of DIT-Tech into existing workflows or management practices.

Numerous participants noted that management failed to effectively communicate the benefits and rationale for using DIT-Tech and thus, were unsure as to why they were instructed to use it. Moreover, some participants raised concerns about the sustainability of new interventions, and felt it was not worth incorporating the tablets into practice if their use would be discontinued in the near future. This contributed to a reluctance to fully embrace and integrate the technology into daily practices.

Staff and care partners participants noted that they did not have privacy concerns with tablet use and they were appreciative of management concerns for privacy when using Skype. All research sites were provided signs to post on room doors and within resident rooms to notify those entering that video streaming may be in progress. One participant explained:

*"There is a notice on the wall, 'Attention all staff. When providing personal care please ensure a resident's tablet is placed out of view as Skype video calls may occur at any time. Thank you.' That was appreciated to see that"* (ID-03, Staff, Female, 58).

However, some staff did have concerns about being watched as they worked:

*"They definitely are more aware and some are more comfortable with it than others because when I've called, if they come in whatever, bring my parents a snack, most of them will just say hello to me. Whereas some try to run away because they don't want to be in the back. I think for some people, there is some confusion about it recording, versus just being a video call"* (ID G-03, Staff, Female, 53).

## Discussion

This qualitative study, using the FITT framework, explored the experience of residents, care partners and staff in the implementation of DIT-Tech as part of a research study. The identified themes revolved around staff attitudes, management support, and resident and care partner interactions. Some residents used the tablets independently, while the personalized content was noted as beneficial by care partners. This contrasts with studies focused on shared technology in facilities[26].

The primary outcomes of this intervention [17], revealed that there was a large variability in tablet use across participants, with a median of 7 minutes per day (with an interquartile range of 27) throughout the study [17]. Higher tablet usage was linked to factors such as younger age, better cognitive function, no hearing impairments, and having a care partner living farther away [17]. However, there were no measurable improvements in quality of life, recreational activities, or relational closeness based on quantitative assessments. These findings are enriched by the qualitative insights that motivation among staff and care partners was a critical facilitator of tablet use. The ability of DIT-Tech to offer personalized experiences tailored to residents' preferences was perceived as enhancing cognitive stimulation, social engagement, and quality of life. Additionally, the qualitative emphasis on perceived, short-term benefits—such as moments of engagement or autonomy—points to a possible mismatch between the outcomes valued by participants and the metrics used in the quantitative evaluation. However, resistance emerged as a substantial barrier, driven by staff concerns about increased workloads, technological literacy gaps, and skepticism regarding older adults' capacity or willingness to use tablets. The study highlights a contrast between the initial enthusiasm and subsequent challenges encountered during implementation.

Motivation among staff and family members was underpinned by the potential of DIT-Tech to supplement caregiving and recreation efforts, with personalized features supporting meaningful engagement. The relationship between motivation and effective implementation underscores the importance of prioritizing and harnessing stakeholder motivation when implementing technology in LTCH. However, implementation does not equate to adoption as technology can be offered or rolled out but not adopted, (i.e., incorporated into an individual's daily life) [27,28]. However, resistance, fueled by past negative experiences with technology and limited resources for training and troubleshooting, dampened the broader adoption of the technology. Unlike residents and family members who valued personalization, many staff viewed the process of customizing devices as burdensome, particularly in the context of existing staffing shortages. These findings underscore the importance of addressing structural barriers, such as providing adequate training and technological support, to reduce resistance and enhance adoption.

One of the most significant benefits identified for residents was the tablet's ability to enhance residents' engagement, by facilitating cognitive stimulation in the form of engagement, social interaction with relatives and staff, and enjoyment of movies and other activities. The high degree of personalization allowed staff and care partners to tailor activities, communication methods, and content to individual residents' interests and combat the sense of loneliness [29]. This approach may prompt questions related to equity concerns, given that numerous LTCH residents lack sufficient access to internet-enabled technologies or the financial means to acquire them, raising questions about fair and inclusive access [30]. However, tablets tend to be a cost-effective technology.

The Technology Acceptance Model (TAM), was proposed to explain the criteria for likely adoption of a new technology. The TAM explains how individuals perceive the usefulness and benefits of a technology (referred to as 'perceived usefulness'), and their intention and willingness to use it (known as 'behavioral intention') [31]. The Senior Technology Acceptance Model (STAM) extended this to address factors relating specifically to aging, including physical and cognitive changes and computer self-efficacy [32]. However, challenges like material and human resource constraints, can intensify reluctance towards adopting new technologies [33]. Future research is needed to understand LTCH residents' knowledge and familiarity with tablets and their willingness and interest in adopting them. This should be undertaken alongside developing strategies to promote and sustain interest and motivation among key stakeholders [34], which also address and alleviate concerns stemming from prior negative experiences through educational initiatives and integration of technology into care practice [35]. In addition, implementation strategies must tackle ageist beliefs that older people, particularly LTCH residents, are not interested or able to use touchscreen technology. This is part of an ongoing need to address age-related biases relating to technology use within healthcare contexts, to increase effective interventions targeting older adults [36]. Encouraging a collaborative approach, where stakeholders can share their experiences and best practices, might also help to normalize the use of personalized technology and older adults using technology [37].

Our findings suggest that comprehensive strategies are needed to ensure the successful integration of DIT-Tech and other personal tablet use into LTCHs. Organizations must effectively communicate the benefits and rationale behind technology implementation to support adoption and sustainability [38]. Future studies could investigate strategies for overcoming the challenges of interdepartmental communication and collaboration, focusing on how to create more effective workflows that enhance the integration of technology into daily practices. Formal and informal technical roles are essential in assisting staff and care partners, reducing downtime and frustration [39], and empowering users to explore tablet functionalities and improve their skills and familiarity with the device, enabling their exploration of the technology and developing their skills and competence [40]. Training for both staff and residents is crucial for the effective utilization of technology [8], and customizing interventions to suit the unique needs and preferences of older adults in long-term care settings is essential for their acceptance and sustained use [41]. Research could examine the long-term sustainability of remote-access technology, including the financial and logistical considerations necessary to maintain these systems over time. Finally, future studies might assess the role of training and support for staff and care partners to ensure that the full potential of these technologies is realized, particularly in terms of ease of use and optimal engagement with residents. By

following LTCHs over extended periods, researchers can also identify trends in social isolation, mental health outcomes, and the evolving integration of technology into care practices.

## Study strengths and limitations

By utilizing semi-structured interviews, the study was able to gather detailed perspectives from a variety of participants—staff, residents, and care partners—allowing for a nuanced understanding of the experiences and challenges associated with the adoption of the DIT-Tech tablet. This approach enabled the exploration of diverse viewpoints and provided deep insights into the contextual factors influencing the implementation of technology in LTCHs. The use of the FITT framework provided a structured lens through which to assess the interaction between users, technology, and clinical tasks. This framework guided the interview process, allowing for a comprehensive exploration of the factors influencing adoption and usability from multiple angles, including individual attitudes, technological features, and organizational contexts. However, the study's reliance on varied technology exposure during interviews limited some participants' detailed feedback on specific features. Recruitment through LTCH site leads might introduce biases. The study primarily relied on dyad interviews with care partners and residents, which may have restricted the depth of insights from residents. Given that many residents echoed their care partners' sentiments, the findings may not fully represent the unique perspectives of residents regarding their experiences with the DIT-Tech tablets. As no individual interviews were conducted with residents, there is a missed opportunity to capture their independent thoughts and feelings about the technology, that future studies can employ. The geographic focus on Ontario also limits the study's applicability to regions with different long-term care policies, resource availability, and healthcare systems. Lastly, given the evolving nature of LTCH environments and resident needs, the study may not fully reflect the long-term sustainability or broader impact of the intervention over time.

## Conclusion

There's an ongoing necessity for innovative strategies supporting the mental well-being of LTCH residents, especially in times of isolation. DIT-Tech introduced a technology-based intervention across seven LTCHs in Ontario, revealing insights into adoption challenges. Staff commitment significantly aided its success, while resistance, skepticism, and increased workload posed hurdles. Integrating tablets into workflows and robust tech support facilitated adoption. The study highlights the need for comprehensive guidelines to support tablet-based interventions in LTCHs. Insights from this study offer valuable lessons for technology-driven initiatives in long-term care. Future research should explore sustainability factors, organizational alignment, and enduring integration into care environments.

## Supporting Information

**S1 Appendix. Pre- and post-Interview guide sample questions.**
(DOCX)

## Author contributions

**Conceptualization:** Kristina M Kokorelias, Alisa Grigorovich, Arlene Astell, Josephine McMurray, Andrea Iaboni.

**Data curation:** Kristina M Kokorelias, Alisa Grigorovich, Teresa D'Elia, Arlene Astell, Josephine McMurray, Andrea Iaboni.

**Formal analysis:** Kristina M Kokorelias, Alisa Grigorovich, Teresa D'Elia, Arlene Astell, Josephine McMurray, Andrea Iaboni.

**Funding acquisition:** Kristina M Kokorelias, Alisa Grigorovich, Arlene Astell, Josephine McMurray, Andrea Iaboni.

**Investigation:** Kristina M Kokorelias, Alisa Grigorovich, Teresa D'Elia, Arlene Astell, Josephine McMurray, Andrea Iaboni.

**Methodology:** Kristina M Kokorelias, Alisa Grigorovich, Arlene Astell, Josephine McMurray, Andrea Iaboni.

**Project administration:** Kristina M Kokorelias, Alisa Grigorovich, Arlene Astell, Andrea Iaboni.

**Resources:** Andrea Iaboni.

**Software:** Andrea Iaboni.

**Supervision:** Kristina M Kokorelias, Alisa Grigorovich, Andrea Iaboni.

**Validation:** Kristina M Kokorelias, Andrea Iaboni.

**Visualization:** Kristina M Kokorelias, Andrea Iaboni.

**Writing – original draft:** Kristina M Kokorelias.

**Writing – review & editing:** Alisa Grigorovich, Teresa D'Elia, Arlene Astell, Josephine McMurray, Andrea Iaboni.

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
