## [Decision Letter · Decision Letter 0]

15 Oct 2024

PONE-D-24-15124Staff, Resident and Care Partner Perceptions on the Use of a Personalized Tablet to Mitigate the Impact of Isolation in Long-Term Care ResidentsPLOS ONE

Dear Dr. Iaboni,

Thank you for submitting your manuscript to PLOS ONE. After careful consideration, we feel that it has merit but does not fully meet PLOS ONE’s publication criteria as it currently stands. Therefore, we invite you to submit a revised version of the manuscript that addresses the points raised during the review process.The reviewers have highlighted a number of important aspects for improvement, e.g. considering transparent reporting.

We look forward to receiving your revised manuscript.

Kind regards,

Sascha Köpke

Academic Editor

PLOS ONE

“This research was funded by a Province of Ontario Rapid Access COVID-19 research grant.  It was also supported by the AGE-WELL Network of Centres of Excellent (NCE) and the Walter and Maria Schroeder Institute for Brain Innovation and Recovery.”

5. We note that this data set consists of interview transcripts. Can you please confirm that all participants gave consent for interview transcript to be published?

If they DID provide consent for these transcripts to be published, please also confirm that the transcripts do not contain any potentially identifying information (or let us know if the participants consented to having their personal details published and made publicly available). We consider the following details to be identifying information:

- Names, nicknames, and initials

- Age more specific than round numbers

- GPS coordinates, physical addresses, IP addresses, email addresses

- Information in small sample sizes (e.g. 40 students from X class in X year at X university)

- Specific dates (e.g. visit dates, interview dates)

- ID numbers

Or, if the participants DID NOT provide consent for these transcripts to be published:

- Provide a de-identified version of the data or excerpts of interview responses

- Provide information regarding how these transcripts can be accessed by researchers who meet the criteria for access to confidential data, including:

a) the grounds for restriction

b) the name of the ethics committee, Institutional Review Board, or third-party organization that is imposing sharing restrictions on the data

c) a non-author, institutional point of contact that is able to field data access queries, in the interest of maintaining long-term data accessibility.

d) Any relevant data set names, URLs, DOIs, etc. that an independent researcher would need in order to request your minimal data set.

For further information on sharing data that contains sensitive participant information, please see: https://journals.plos.org/plosone/s/data-availability#loc-human-research-participant-data-and-other-sensitive-data

If there are ethical, legal, or third-party restrictions upon your dataset, you must provide all of the following details (https://journals.plos.org/plosone/s/data-availability#loc-acceptable-data-access-restrictions):

1. A complete description of the dataset

2. The nature of the restrictions upon the data (ethical, legal, or owned by a third party) and the reasoning behind them

3. The full name of the body imposing the restrictions upon your dataset (ethics committee, institution, data access committee, etc)

4. If the data are owned by a third party, confirmation of whether the authors received any special privileges in accessing the data that other researchers would not have

5. Direct, non-author contact information (preferably email) for the body imposing the restrictions upon the data, to which data access requests can be sent.

Reviewers' comments:

Reviewer's Responses to Questions

**Comments to the Author**

1. Is the manuscript technically sound, and do the data support the conclusions?

Reviewer #1: Yes

Reviewer #2: Yes

2. Has the statistical analysis been performed appropriately and rigorously? 

Reviewer #1: N/A

Reviewer #2: N/A

3. Have the authors made all data underlying the findings in their manuscript fully available?

Reviewer #1: Yes

Reviewer #2: Yes

4. Is the manuscript presented in an intelligible fashion and written in standard English?

Reviewer #1: Yes

Reviewer #2: Yes

5. Review Comments to the Author

Reviewer #1: The paper describes experiences with a tech intervention, DIT-Tech, which aims to enhance LTCH residents’ wellbeing by addressing social isolation. The presentation of the findings is clear and interesting. However, some details about the research are missing; this information may have implications for the interpretation of the findings and/or the limitations of the research.

Under “Setting and intervention”, it states that 7 “diverse” LTCHs participated in the research but no details are provided. How were the homes diverse? How were LTCHs recruited?

No information is provided on the characteristics of residents who used the tablets. Given the high proportion of LTCH residents living with dementia, it would be particularly important to know what proportion of residents who used the tablets had dementia. This information would provide important context for interpreting the findings. If this information is not available, the potential impacts should be discussed in the limitations section. It would also be important to know if any of the residents who were interviewed had dementia and what proportion of care partners had relatives with dementia. If there is not good representation of residents with dementia and/or care partners of relatives have dementia, then the impacts of this should be discussed in the discussion and/or the limitations section. For example, future research directions are discussed on page 25 but there’s no mention of ensuring research includes the perspectives and voices of people living with dementia. Their use, and perceptions of DIT-Tech may differ from residents without dementia.

No residents were interviewed individually – all were part of interviews with care partners. Were residents active participants in the interviews? On page 9 it states that “resident quotes were largely omitted from the excerpts”, stating that there was agreement between residents’ and care partners’ comments. Why were most resident quotes (vs. care partner quotes) omitted? The tablets are intended for LTCH residents, and the aim of the study is to examine perceptions of the tablets. Given this, the perceptions of residents and staff would be the most relevant. Yet the manuscript only includes one quote from a resident. Please consider the use of additional resident quotes. Also discuss, as appropriate, the limited input/data from residents in the discussion or limitations sections.

Was each tablet provided to a single resident for use? How did the personalization process work? Was it based on information provided by residents? care partners? And was this done through discussion, survey, interview or some other approach? If a tablet was provided to another resident, would it be re-programmed? Was the research team responsible for individualizing the tablet, or the staff?

For what period of time / how frequently were the tablets used by residents and relatives of care partners? Also, what role did interviewed staff members play in implementing/supporting residents’ use of the tablets. This information would provide important context for interpreting the findings.

The limitations identified in the limitations section should be expanded upon and the implication(s) of each limitation discussed.

Minor comments:

Under “Design” – there is only mention of understanding “staff” perspectives, not residents and care partners.

Table 1 is incomplete. Information regarding ethnicity is said to be “pending”.

Some rows in the tables include document line numbers.

DIT-Tech is sometimes written with a capital “I” and sometimes with a lower case “I”.

Line 454 – should this say “…differing from shared technology studies”?

Line 469 – there is a reference to “leisure wellbeing” but elsewhere well-being (hyphenated) was usually described as mental well-being.

Reviewer #2: Thank you for the opportunity to review this manuscript which aimed to develop an understanding of the staff, resident and care partner experiences, including barriers and facilitators, to the adoption of remote-access personal tablets in long-term care home.

Mayor Revision

• Abstract: Please explain the abbreviation LTCH.

• Setting and intervention: The description of the intervention and the setting is not very detailed. Please describe some typical characteristics of long-term care homes for an international readership (e.g. ownership, size related to resident places, implemented care concepts, qualification levels of staff).

• Setting and intervention: The intervention is a complex intervention. Please use the recommendations of the TiDier Guideline to describe the intervention. For example, it is currently unclear who used the tablets, when, how often and under what circumstances. It is also unclear what role and qualification the DIT Tech Coordinator had?

https://www.equator-network.org/reporting-guidelines/tidier/#:~:text=The%20TIDieR%20Checklist%20is%20available%20to%20download%20as%20a%20PDF

• Participants and Recruitment: What does tablet usage level mean? What was the distribution of the tablet usage level.? Please present the tablet usage level as a result in terms of the degree of implementation of your intervention.

• Resident participant characteristics: What were the reasons for the isolation of the residents? To what extent did the duration of isolation have an influence on the selection of residents?

• Staff participant characteristics: Please describe the qualifications and roles of the employees so that an international readership can categorise them (e.g. using an international qualification framework)

• Discussion: Please reflect on future scientific steps necessary for the development and evaluation of the analysed intervention? In particular, please reflect on the necessity of investigating the effectiveness in corresponding study designs.

6. PLOS authors have the option to publish the peer review history of their article (what does this mean? ). If published, this will include your full peer review and any attached files.

**Do you want your identity to be public for this peer review?** For information about this choice, including consent withdrawal, please see our Privacy Policy .

Reviewer #1: No

Reviewer #2: No

---

## [Author Response · Author response to Decision Letter 1]

25 Nov 2024

RE: PONE-D-24-15124- Staff, Resident and Care Partner Perceptions on the Use of a Personalized Tablet to Mitigate the Impact of Isolation in Long-Term Care Residents

Dear Prof. Köpke,

Thank you for your interest in our manuscript and for the constructive feedback.

Below we outline the changes we have made to the manuscript, as per the feedback from the editor and reviewer. We have created an itemized list of all the reviewer comments and our responses. We have presented changes in the manuscript using ‘track’ changes

Andrea Iaboni, MD DPhil FRCPC

Geriatric Psychiatrist and Medical Lead, Seniors Mental Health, University Health Network

Senior Scientist, KITE, Toronto Rehab Research Institute

Associate Professor, Dept. of Psychiatry, University of Toronto

Comments to the Author

Reviewer #1:

1. Reviewer Comments to Authors: The paper describes experiences with a tech intervention, DIT-Tech, which aims to enhance LTCH residents’ wellbeing by addressing social isolation. The presentation of the findings is clear and interesting. However, some details about the research are missing; this information may have implications for the interpretation of the findings and/or the limitations of the research.

Response to Reviewers: Thank you for your feedback and for highlighting the clarity and interest in our presentation of DIT-Tech. To address your request for further details, we have expanded on the research design, participant selection, and the contextual factors that may influence the interpretation of findings. We have also added further discussion on the limitations of the study to ensure a more comprehensive understanding of the results and their implications.

2. Reviewer Comments to Authors: Under “Setting and intervention”, it states that 7 “diverse” LTCHs participated in the research but no details are provided. How were the homes diverse? How were LTCHs recruited?

Response to Reviewers: The term “diverse” in reference to the 7 participating LTCHs reflects differences in their geographical locations, resident populations, and available resources. We now write:

The recruitment of LTCHs involved a strategic approach, wherein potential sites were identified based on existing collaborations with the research team and their willingness to participate. The long-term care homes represented urban and rural settings and cared for residents of various socioeconomic backgrounds, and with varied resident needs. The diversity of the LTCHs was characterized in terms of representation of different regions of Ontario and size of community. Of the 7 homes, 2 were located in a large city, 3 were located in different small cities (50-150,000 people), and 2 were located in small towns (5-15,000 people). 5 were operated by not-for-profit organizations and 2 were operated by a municipality. In terms of size of the homes, they ranged between 106-228 beds per home, with an average size of 142 beds. This diversity was crucial to capture a broad spectrum of perspectives and experiences regarding the implementation of the DIT-Tech tablet intervention.

3. Reviewer Comments to Authors: No information is provided on the characteristics of residents who used the tablets. Given the high proportion of LTCH residents living with dementia, it would be particularly important to know what proportion of residents who used the tablets had dementia. This information would provide important context for interpreting the findings. If this information is not available, the potential impacts should be discussed in the limitations section. It would also be important to know if any of the residents who were interviewed had dementia and what proportion of care partners had relatives with dementia. If there is not good representation of residents with dementia and/or care partners of relatives have dementia, then the impacts of this should be discussed in the discussion and/or the limitations section. For example, future research directions are discussed on page 25 but there’s no mention of ensuring research includes the perspectives and voices of people living with dementia. Their use, and perceptions of DIT-Tech may differ from residents without dementia.

Response to Reviewers: The demographic characteristics of the full study sample are available in the publication of the primary outcomes of the DIT-Tech study, now referenced in the text.

In this paper, we have provided the demographic information of those residents and care partners involved in the interviews as part of this qualitative aim. It was an omission that we did not include information about the cognitive abilities of resident participants. Diagnosis of dementia is not accurately captured in the health administrative data available to us for this study, but all LTC residents receive a resident assessment of cognitive function. We now include the Cognitive Performance Scale (CPS) of interview participants in Table 2. Two participants did not have cognitive impairment (CPS score 0), one had mild impairment and 4 had moderate impairment.

4. Reviewer Comments to Authors: No residents were interviewed individually – all were part of interviews with care partners. Were residents active participants in the interviews? On page 9 it states that “resident quotes were largely omitted from the excerpts”, stating that there was agreement between residents’ and care partners’ comments. Why were most resident quotes (vs. care partner quotes) omitted? The tablets are intended for LTCH residents, and the aim of the study is to examine perceptions of the tablets. Given this, the perceptions of residents and staff would be the most relevant. Yet the manuscript only includes one quote from a resident. Please consider the use of additional resident quotes. Also discuss, as appropriate, the limited input/data from residents in the discussion or limitations sections.

Response to Reviewers: Thank you for your feedback regarding the engagement of residents in the interviews. We acknowledge that residents were not often engaged in the discussions and frequently echoed the sentiments expressed by their care partners, providing brief replies. We had noted “Notably, resident quotes were largely omitted from the excerpts, as they frequently echoed sentiments expressed by their care partners with brief replies. However, there were no instances of contradictory responses between dyad participants.” Furthermore, we now write:

The study primarily relied on dyad interviews with care partners and residents, which may have restricted the depth of insights from residents. Given that many residents echoed their care partners' sentiments, the findings may not fully represent the unique perspectives of residents regarding their experiences with the DIT-Tech tablets. As no individual interviews were conducted with residents, there is a missed opportunity to capture their independent thoughts and feelings about the technology, that future studies can employ.

5. Reviewer Comments to Authors: Was each tablet provided to a single resident for use? How did the personalization process work? Was it based on information provided by residents? care partners? And was this done through discussion, survey, interview or some other approach? If a tablet was provided to another resident, would it be re-programmed? Was the research team responsible for individualizing the tablet, or the staff? For what period of time / how frequently were the tablets used by residents and relatives of care partners? Also, what role did interviewed staff members play in implementing/supporting residents’ use of the tablets. This information would provide important context for interpreting the findings.

Response to Reviewers: Thank you for your comments. We now write:

Briefly, tablets were provided to a single resident for their personal use, ensuring that the technology was tailored to their individual needs and preferences. The personalization of each tablet was a collaborative effort that involved information gathered from both residents and their care partners. This process included discussions with care staff aimed at understanding the residents’ interests, preferred communication methods, and any specific applications that would enhance their engagement. If a tablet was to be reassigned to another resident, it would be reprogrammed to reflect the new resident’s preferences and requirements. The responsibility for individualizing the tablets primarily lay with the staff, who worked closely with the research team to implement the necessary customizations based on resident input and care partner feedback. The staff members interviewed played a significant role in supporting residents’ use of the tablets. They assisted with training, troubleshooting, and ongoing encouragement for residents to engage with the technology.

6. Reviewer Comments to Authors: The limitations identified in the limitations section should be expanded upon and the implication(s) of each limitation discussed.

Response to Reviewers: We have expanded our limitations. This section now reads as:

However, the study's reliance on varied technology exposure during interviews limited some participants' detailed feedback on specific features. Recruitment through LTCH site leads might introduce biases. The study primarily relied on dyad interviews with care partners and residents, which may have restricted the depth of insights from residents. Given that many residents echoed their care partners' sentiments, the findings may not fully represent the unique perspectives of residents regarding their experiences with the DIT-Tech tablets. As no individual interviews were conducted with residents, there is a missed opportunity to capture their independent thoughts and feelings about the technology, that future studies can employ. The geographic focus on Ontario also limits the study's applicability to regions with different long-term care policies, resource availability, and healthcare systems. Lastly, given the evolving nature of LTCH environments and resident needs, the study may not fully reflect the long-term sustainability or broader impact of the intervention over time.

7. Reviewer Comments to Authors: Under “Design” – there is only mention of understanding “staff” perspectives, not residents and care partners.

Response to Reviewers: We now write “This qualitative descriptive study (Sandelowski, 2010), used purposive sampling to recruit staff members, residents and care partners from multiple long-term care homes (LTCH) who were directly involved in the implementation of DIT-Tech”.

8. Reviewer Comments to Authors: Table 1 is incomplete. Information regarding ethnicity is said to be “pending”.

Response to Reviewers: Thank you for identifying this omission in Table 1This row was added incorrectly as ethnicity data was not captured for care partners. It has now been removed.

9. Reviewer Comments to Authors: Some rows in the tables include document line numbers.

Response to Reviewers: We appreciate you bringing this to our attention, and we will ensure that the tables are revised to remove any extraneous line numbers in the final version of the manuscript.

10. Reviewer Comments to Authors: DIT-Tech is sometimes written with a capital “I” and sometimes with a lower case “I”.

Response to Reviewers: Thank you for pointing out the inconsistency in the capitalization of "DIT-Tech." We will ensure that the term is uniformly presented with a capital “I” throughout the manuscript to maintain clarity and consistency. Your feedback is greatly appreciated.

11. Reviewer Comments to Authors: Line 454 – should this say “…differing from shared technology studies”?

Response to Reviewers: Thank you for your observation regarding Line 454. It should indeed read “differing from shared technology studies.”

12. Reviewer Comments to Authors: Line 469 – there is a reference to “leisure wellbeing” but elsewhere well-being (hyphenated) was usually described as mental well-being.

Response to Reviewers: Thank you for pointing out the inconsistency regarding the term "leisure wellbeing." We have revised Line 469 to ensure that it aligns with the hyphenated "mental well-being" used elsewhere in the manuscript.

Reviewer #2

13. Reviewer Comments to Authors: Thank you for the opportunity to review this manuscript which aimed to develop an understanding of the staff, resident and care partner experiences, including barriers and facilitators, to the adoption of remote-access personal tablets in long-term care home.

Response to Reviewers: Thank you for your thoughtful review of our manuscript. We appreciate your recognition of the study's aim to understand the experiences of staff, residents, and care partners regarding the adoption of remote-access personal tablets in long-term care homes.

14. Reviewer Comments to Authors: Abstract: Please explain the abbreviation LTCH.

Response to Reviewers: We now write “This study aimed to develop an understanding of the staff, resident and care partner experiences, including barriers and facilitators, to the adoption of remote-access personal tablets in long-term care home (LTCH).”

15. Reviewer Comments to Authors: Setting and intervention: The description of the intervention and the setting is not very detailed. Please describe some typical characteristics of long-term care homes for an international readership (e.g. ownership, size related to resident places, implemented care concepts, qualification levels of staff). The intervention is a complex intervention. Please use the recommendations of the TiDier Guideline to describe the intervention. For example, it is currently unclear who used the tablets, when, how often and under what circumstances. It is also unclear what role and qualification the DIT Tech Coordinator had?

https://www.equator-network.org/reporting-guidelines/tidier/#:~:text=The%20TIDieR%20Checklist%20is%20available%20to%20download%20as%20a%20PDF

Response to Reviewers: Thank you for your comments. We have added these additional details as well as referencing a publication which provides further details about the methods and results of the primary aim of this study.

We now write:

The recruitment of LTCHs involved a strategic approach, wherein potential sites were identified based on existing collaborations with the research team and their willingness to participate. The long-term care homes represented urban and rural settings and cared for residents of various socioeconomic backgrounds, and with varied resident needs. The diversity of the LTCHs was characterized in terms of representation of different regions of Ontario and size of community. Of the 7 homes, 2 were located in a large city, 3 were located in different small cities (50-150,000 people), and 2 were located in small towns (5-15,000 people). 5 were operated by not-for-profit organizations and 2 were operated by a municipality. In terms of size of the homes, they ranged between 106-228 beds per home, with an average size of 142 beds. This diversity was crucial to capture a broad spectrum of perspectives and experiences regarding the implementation of the DIT-Tech tablet intervention. These Android tablets, property of the LTCHs, feature standard apps and Skype. Key features include remote control access, direct family calls via auto-answer on Skype, and personalized setups for residents (Astell et al., 2024).

Full study details on the implementation of the tablet are available in the larger study’s manuscript (Astell et al., 2024).

Briefly, tablets were provided to a single resident for their personal use, ensuring that the technology was tailored to their individual needs and preferences. The personalization of each tablet was a collaborative effort that involved information gathered from both residents and their care partners. This process included discussions with care staff aimed at understanding the residents’ interests, preferred communication methods, and any specific applications that would enhance their engagement. If a tablet was to be reassigned to another resident, it would be reprogrammed to reflect the new resident’s preferences and requirements. The responsibility for individualizing the tablets primarily lay wit

---

## [Editor Report · Decision Letter 1]

27 Jan 2025

Staff, Resident and Care Partner Perceptions on the Use of a Personalized Tablet to Mitigate the Impact of Isolation in Long-Term Care Residents

PONE-D-24-15124R1

Dear Dr. Iaboni,

Sorry for the delay, due to reviewers who required more time than usual!

We’re pleased to inform you that your manuscript has been judged scientifically suitable for publication and will be formally accepted for publication once it meets all outstanding technical requirements.

Kind regards,

Sascha Köpke

Academic Editor

PLOS ONE

---

## [Editor Report · Acceptance letter]

PONE-D-24-15124R1

PLOS ONE

Dear Dr. Iaboni,

I'm pleased to inform you that your manuscript has been deemed suitable for publication in PLOS ONE. Congratulations! Your manuscript is now being handed over to our production team.

Kind regards,

on behalf of

Professor Sascha Köpke

Academic Editor

PLOS ONE